# Alias-Free ViT:
# Fractional Shift Invariance via Linear Attention

**Hagay Michaeli**
Technion
Haifa, Israel
hagaymi@campus.technion.ac.il

**Daniel Soudry**
Technion
Haifa, Israel
daniel.soudry@gmail.com

## Abstract

Transformers have emerged as a competitive alternative to convnets in vision tasks, yet they lack the architectural inductive bias of convnets, which may hinder their potential performance. Specifically, Vision Transformers (ViTs) are not translation-invariant and are more sensitive to minor image translations than standard convnets. Previous studies have shown, however, that convnets are also not perfectly shift-invariant, due to aliasing in downsampling and nonlinear layers. Consequently, anti-aliasing approaches have been proposed to certify convnets translation robustness. Building on this line of work, we propose an Alias-Free ViT, which combines two main components. First, it uses alias-free downsampling and nonlinearities. Second, it uses linear cross-covariance attention that is shift-equivariant to both integer and fractional translations, enabling a shift-invariant global representation. Our model maintains competitive performance in image classification and outperforms similar-sized models in terms of robustness to adversarial translations.[1]

## 1 Introduction

Transformers, primarily designed for language modeling [58], have become dominant in vision tasks [26, 34]. Since they were originally designed for sequential data, their underlying attention mechanism is not sensitive to the locality of information in visual data. As a result, Vision Transformers (ViTs) exhibit a lack of shift-invariance, a shortcoming that becomes evident in cases where small image translations lead to significant deviations in output [25, 51].

To mitigate this gap, many studies have been conducted on the integration of convolutional priors, such as spatial hierarchy and shift-invariance, into ViT architectures. For example, approaches include hierarchical patch merging [39], the incorporation of convolutional layers [61], and the design of relative positional encodings [62]. Furthermore, some works have drawn parallels between self-attention and dynamic convolutions [1, 9, 27], motivating reinterpretations of the attention mechanism through a convolutional lens.

Despite being more spatially aware than transformers, convnets are not perfectly shift-invariant due to aliasing introduced by strided convolutions and pooling layers [4, 67]. This has led to a line of research that aims to restore shift-invariance, by methods including anti-aliasing filters [21, 23, 32, 40, 67, 71] and adaptive sampling techniques [6, 46]. Building on these advances, recent works have adapted such methods for transformer architectures. For example, Adaptive Polyphase Sampling (APS) has been employed to achieve cyclic shift-equivariance in ViTs [13, 47]. However, despite the latter approach efficiently guaranteeing shift-invariance for integer pixel cyclic shifts, it falls short in fractional (i.e., sub-pixel) shifts and "standard" shifts (i.e. imitating camera translation) [11, 40, 51].

---

[1]Our code is available at github.com/hmichaeli/alias_free_vit.

39th Conference on Neural Information Processing Systems (NeurIPS 2025).

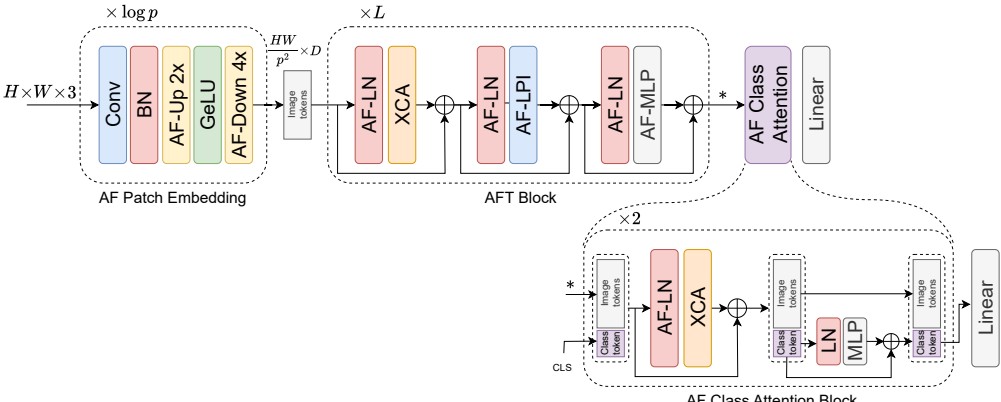

Figure 1: **Overview of the Alias-Free Vision Transformer (AFT) architecture.** The input image is first processed by an alias-free patch embedding module composed of convolutional layers (Conv), batch normalization (BN), and alias-free activation, composed of upsampling (AF-Up), GELU and downsampling (AF-Down). The result is reshaped to a token matrix form and fed through $L$ Alias-Free Transformer blocks, each consisting of alias-free layer normalization (AF-LN), cross-covariance attention (XCA), alias-free local patch interaction (AF-LPI), and alias-free MLP (AF-MLP) layers, interconnected by residual connections. The result is concatenated with a learnable class token embedding and fed into two Alias-Free Class Attention blocks composed of an XCA layer and an MLP applied on the class token. The final representation is the updated class token, which is fed into a final linear classifier. Detailed explanations of each component are provided in Section 3.

As aliasing is primarily related to downsampling layers, which are not frequently used in ViTs, only few studies have been conducted on integrating aliasing-reduction techniques to achieve shift-invariance in ViTs. Qian et al. [42] propose plugging a low-pass filter post self-attention to reduce aliasing, however, this only provides a partial solution, as it does not resolve the inherent lack of shift-equivariance in self-attention and aliasing in other nonlinearities. Recent works study aliasing in latent diffusion models [2, 64, 70] that typically include attention layers, in order to improve their consistency i.e. in video generation. However, these works do not address the main issue in the self-attention operation. A similar problem may also hinder transformer neural operators which have recently become popular [28, 38, 52], as aliasing has been shown to be related to discretization errors [5, 44, 69].

Another emerging direction focuses on linear and softmax-free attention mechanisms, initially proposed to reduce the quadratic complexity of standard attention in large language models (LLMs) [7, 33, 59]. In the vision domain, models such as XCiT [16] and SimA [35] demonstrate that alternative attention formulations, e.g., using cross-covariance or linear attention, can maintain competitive performance without directly computing a full attention map. Beyond improved efficiency, we now show that such mechanisms enable designing a transformer-based architecture that is shift-equivariant, similar to convnets.

**Contributions.**    In this paper

- We present in Section 2 a certain class of shift-equivariant attention layers (SEA), including linear attention and cross-covariance attention, which is useful for vision tasks.

- We design in Section 3 a shift-invariant, alias-free Vision Transformer (AFT) using cross-covariance attention and alias-free nonlinearities and show it has a competitive performance in image classification.

- We show in Section 4 that the AFT is robust ($\sim 99\%$ consistency) to fractional cyclic shifts, and significantly more robust to practical translations than other similar models, albeit with increased computational overhead due to the alias-free components.

## 2 Methods

### 2.1 Preliminaries: the Vision Transformer

The Vision Transformer (ViT) [15] transfers the Transformer architecture [58] from text to images by interpreting an image as a sequence of visual tokens. Given an input $x \in \mathbb{R}^{C \times H \times W}$, the image is partitioned into $N = \frac{HW}{p^2}$ non-overlapping patches of resolution $p \times p$. Each patch is flattened and projected by a learned linear layer into a $D$-dimensional embedding, forming $X \in \mathbb{R}^{N \times D}$. For classification, a learnable "class" token is prepended to $X$ and later serves as a global representation, propagated into the classification module. Learnable *absolute* positional encodings $\mathcal{P} \in \mathbb{R}^{(N+1) \times D}$ are then added to compensate for the permutation-invariance of self-attention.

The resulting sequence $\tilde{X} = X + \mathcal{P}$ is processed by $L$ identical Transformer encoder blocks. Each block is composed of a Multi-Head Self-Attention (MHSA) module with a two-layer Feed-Forward Network (FFN), interleaved with Layer Normalization (LN) and residual connections:

$$\hat{X}^{(\ell)} = \tilde{X}^{(\ell-1)} + \text{MHSA}\left(\text{LN}\left(\tilde{X}^{(\ell-1)}\right)\right), \tag{1}$$

$$\tilde{X}^{(\ell)} = \hat{X}^{(\ell)} + \text{FFN}\left(\text{LN}\left(\hat{X}^{(\ell)}\right)\right), \tag{2}$$

where $\ell = 1, \ldots, L$ and $\tilde{X}^{(0)} = \tilde{X}$.

**Multi-Head Self-Attention.** Each of the $h$ heads linearly projects the input into queries, keys and values, $Q = XW_q$, $K = XW_k$, $V = XW_v$, with $W_q, W_k, W_v \in \mathbb{R}^{D \times d_h}$ and $d_h = D/h$. Self-Attention is then computed as

$$\text{SA}(X) = \text{softmax}\left(QK^\top / \sqrt{d_h}\right) V. \tag{3}$$

The outputs of all heads are concatenated and projected back to $\mathbb{R}^D$ by a final linear layer.

**Feed-Forward Network.** The FFN first expands the embedding dimension to $4D$, applies a GELU activation [30], and projects back:

$$\text{FFN}(X) = W_2 \, \text{GELU}(W_1 X + b_1) + b_2, \tag{4}$$

with $W_1 \in \mathbb{R}^{4D \times D}$ and $W_2 \in \mathbb{R}^{D \times 4D}$.

### 2.2 Linear Attention

Notably, Equation (3) requires computing $QK^\top \in \mathbb{R}^{N \times N}$ explicitly, which has a quadratic cost in the number of tokens. This has motivated many linear complexity variants of self-attention [7, 33, 59]. In vision, SimA removes softmax entirely by maintaining stability using $\ell_1$–normalized $Q$ and $K$ [35]. XCiT mixes channels instead of tokens with cross-covariance attention (XCA) [16],

$$\text{XCA}(X) = V \, \text{softmax}\left(\hat{K}^\top \hat{Q} / \tau\right), \tag{5}$$

where $\hat{Q}, \hat{K}$ are $\ell_2$-normalized along the token dimension and $\tau > 0$ is a learnable temperature.

### 2.3 Alias-Free Vision Transformer

Next, we describe our proposed model, replacing every ViT component that is not shift-equivariant with a modification that restores equivariance. For brevity, the analysis is given for a one-dimensional signal. The same principles can be applied in the two-dimensional case by viewing the sequence of $N$ tokens in a two-dimensional representation, namely $X \in \mathbb{R}^{\frac{H}{p} \times \frac{W}{p} \times D}$.[2]

Similar to Karras et al. [32] and Michaeli et al. [40], we view the input as a discrete sampling of an underlying band-limited continuous signal. The main difference from a standard convnet emerges

---

[2]In practice two-dimensional signals tokens are also stacked into a one-dimensional sequence (formally the row-stack). Nevertheless, shift-equivariance in the two dimensions is still maintained under this representation.

after tokenization: the patch-embedding matrix $X \in \mathbb{R}^{N \times D}$ stores the sequence length $N$ along its first axis, and the embedding dimension $D$ in the second axis. Thus, the spatial and channel roles are swapped compared to convnet feature maps, where the channel index comes first and spatial indices follow. Throughout our analysis, we therefore interpret the $D$ columns of $X$ as $D$ "channels" of a length-$N$ one-dimensional signal. Maintaining shift-equivariance then amounts to ensuring that each column transforms equivariantly under fractional translations in the continuous domain.

**Shift invariance and equivariance.** We reuse the definitions of Michaeli et al. [40]. Let $x[n]$ be a discrete signal, $T$ its sample spacing, and $z(t)$ the unique $\frac{1}{2T}$-band-limited signal with $x[n] = z(nT)$. For any (possibly non-integer) shift $\Delta \in \mathbb{R}$,

$$\tau_\Delta x[n] \; = \; z(nT + \Delta).$$

An operator $f$ is *shift-equivariant* if $f(\tau_\Delta x) = \tau_\Delta f(x)$ and *shift-invariant* if $f(\tau_\Delta x) = f(x)$ for all $x$ and $\Delta$.

In some cases, we may claim that a value is shift-equivariant. This is a slight abuse of the definition to say the overall operator computing this value is shift-equivariant w.r.t. the input signal.

**Patch embedding.** Algebraically, the tokenization process described in Section 2.1 is equivalent to a convolution with kernel size $p$, stride $p$, and $D$ output channels. By separating this stride-$p$ convolution into a stride-1 convolution followed by an alias-free downsampling [21], the composite operator becomes shift-equivariant [32, 40, 67]. However, this requires inserting a single low-pass filter with cut-off $1/p$, which would severely attenuate high-frequency content, especially for large patches. Instead, we employ a convolutional patch-embedding that replaces the single stride-$p$ layer with a short convnet of progressively smaller strides, often used in hybrid models [16, 24, 61, 66]. Here, we can avoid aliasing by plugging a low-pass filter with a larger cut-off before each downsampling layer, and by using an alias-free activation function [32, 40]. This gradual approach still enables the network to learn high-frequency features, despite the anti-aliasing components.

**Positional encoding.** Absolute positional encoding injects the global index of each token and therefore breaks shift-equivariance. Several relative schemes have been proposed that depend only on pairwise offsets and thus preserve shift-equivariance at the token level [8, 39, 50]. These methods, however, do not guarantee equivariance to pixel-level translations, as they cause the patch contents themselves to change. Notably, the convolutional patch embedding already breaks permutation invariance of the tokens, possibly reducing the need for additional positional signals. Moreover, recent studies demonstrate that hybrid transformers can learn effectively without any explicit positional encoding [3, 68]. Consistent with this observation, we find empirically (Section 4.4) that positional encoding may be unnecessary in architectures that include convolutional layers inside the transformer blocks, such as XCiT [16].

**Shift-Equivariant Attention.** We next show a class of attention operations that, by removing the softmax in Equation (3), are shift-equivariant. This primarily includes the linear attention, which, although still less common, is attractive for its lower complexity [33, 54] and yields competitive vision results [35]. Formally,

$$\mathrm{SEA}\left(X\right) \; = \; Qf\left(K^\top V\right), \tag{6}$$

where we keep the existing notation $Q = XW_q, \; K = XW_k, \; V = XW_v$, and let $f : \mathbb{R}^{D \times D} \to \mathbb{R}^{D \times D}$ be an arbitrary function.

We now establish the desired property in three steps.

**Proposition 1.** *$Q, K$ and $V$ are shift-equivariant.*

*Proof.* Each column of $Q$, $K$ or $V$ is a fixed linear combination of the columns of $X$. As the columns of $X$ are merely the channels of the same signals, they translate together, and any linear combination of them is also shift-equivariant.

$\square$

**Proposition 2.** *$f\left(K^\top V\right)$ is shift-invariant.*

*Proof sketch.* Consider the matrix entry $\left(K^\top V\right)_{ij} = K_i^\top V_j$, where $K_i$ and $V_j$ denote columns. By Parseval's theorem,

$$K_i^\top V_j = \frac{1}{2\pi} \int_{-\pi}^{\pi} \hat{K}_i(\omega)\, \hat{V}_j^*(\omega)\, d\omega.$$

A fractional translation by $\tau$ multiplies both Fourier transforms by the same phase factor $e^{j\omega\tau}$, which cancels in the product. Hence, every entry of $K^\top V$ remains unchanged due to a translation (see formal proof in appendix A). An important observation is that since $K^\top V$ is shift-*invariant*, any matrix operation can be applied on it without compromising this property.

**Proposition 3.** $V' = Q f\left(K^\top V\right)$ *is shift-equivariant.*

*Proof.* Column $j$ of $V'$ satisfies

$$V_j' = \sum_{i=1}^{D} Q_i\, f\left(K^\top V\right)_{ij},$$

i.e. a sum of *shift-equivariant* columns $Q_i$ (Proposition 1) scaled by *shift-invariant* coefficients $f\left(K^\top V\right)_{ij}$ (Proposition 2). The resulting column is therefore shift-equivariant. $\qquad\square$

As mentioned above, since $K^\top V$ is shift-*invariant*, any matrix operation $f$ can be applied on it without compromising the overall shift-*equivariance*. By the same argument, $K^\top Q$ is also shift-invariant; thus a row-wise softmax on $K^\top Q$ as used in XCA (Equation (5)) is permissible. Note that in XCA, the columns of $Q$ and $K$ are $\ell_2$-normalized along the token dimension, which as well maintains shift-equivariance [40].

**MLP.** The MLP linear layers apply the same linear transformation on all tokens, maintaining shift-equivariance as in Proposition 1. However, pointwise nonlinearities induce aliasing that breaks fractional shift-equivariance. This can be solved similarly to [40] by an alias-free activation function, which includes upsampling before the nonlinearity and downsampling back after low-pass filtering.

**Layer normalization.** Per-token LayerNorm rescales each column differently and breaks equivariance. The same problem has been addressed by Michaeli et al. [40], by using a global variant, namely

$$\hat{X}_{ij} = \frac{X_{ij} - \mu}{\sqrt{\sigma^2 + \epsilon}}, \quad \mu_i = \tfrac{1}{D}\sum_j X_{ij}, \ \sigma^2 = \tfrac{1}{ND}\sum_{i,j}(X_{ij} - \mu_i)^2. \tag{7}$$

where $\mu$ is computed per token and $\sigma^2$ per layer.

**Class token.** Prepending the "class" token interferes with the signal representation of the columns of the embedding matrix, and breaks shift-equivariance. A simple solution is to instead construct a global representation using global average pooling over the embedding dimension after the last transformer block, i.e.

$$\bar{X} = \frac{1}{N}\sum_{i=1}^{N} X_i^{(L)} \in \mathbb{R}^D, \tag{8}$$

similar to the common approach in convnets and a few other ViTs [24, 39, 65]. This, assuming shift-equivariance is maintained, yields a shift-invariant global representation [40].

**Class Attention.** Some ViTs append the class token only after $L$ patch-only blocks and update it through a few class-attention (CA) layers [10, 16, 56]. These layers usually fix the patch embeddings and update only the class token embedding by attending it to itself and to the frozen patch embeddings. We find that when using SEA after concatenating the class token, Propositions 1 to 3 still hold w.r.t. the patch tokens, and additionally the class token remains shift-invariant (see proof in appendix A). Retaining similarity to the original class attention blocks, we only propagate the class embedding through the MLP, which also prevents aliasing in the nonlinearity. We use this approach instead of global average pooling since we find it performs slightly better.

# 3  Implementation

We implement the Alias-Free Vision Transformer (AFT) based on XCiT architecture [16]. XCiT replaces standard self-attention with cross-covariance attention, described in Equation (5). This, by Proposition 3, preserves shift-equivariance. The remaining modifications focus on eliminating aliasing in the patch-embedding, local patch interaction (LPI), and MLP blocks. The overall model is presented in Figure 1.

**Conv Patch Embedding.**   XCiT patch embedding (PE) module is a sequence of blocks composed of strided convolution, batch normalization, and GELU. Similar to [40], we replace the strided convolution with a stride of 1 and insert an alias-free downsampling at the end of the block, implemented by truncation of high-frequencies in the Fourier domain, using Fast Fourier Transform (FFT) [22, 45]. We replace zero-padding with circular padding. We use alias-free activation functions by wrapping the GELU activation with upsampling and alias-free downsampling layers [32, 40]. In contrast to [40], we find that replacing the GELU with a polynomial activation to get perfect shift-invariance degrades the model performance significantly. Conversely, we find that keeping GELU affects the translation-robustness marginally. We do not add positional encodings and do not prepend a class token at this stage; the class token is appended only before the class attention blocks.

**AFT Block.**   Following the patch embedding, XCiT is composed of a sequence of Blocks, each consisting of three components: cross-covariance attention (XCA), local patch interaction (LPI), and MLP. As argued in Section 2.3, the XCA is already shift-equivariant. The LPI consists of two convolutional layers, batch normalization, and activation layers, and we treat each of them as in the PE, forming an alias-free version we denote AF-LPI. The MLP applies a shared two-layer FFN on each token, and can be viewed as a convolutional layer with kernel size 1. Therefore, the only required modification to form the alias-free variant (AF-MLP) is converting the activation function into an alias-free activation. We replace all LayerNorm instances, applied before XCA, LPI, and MLP, with the alias-free layer norm described in Section 2.3.

**Classification.**   After the last AFT block we append a learnable class token and apply two *AF class attention* blocks. Each block consists of AF-LN, an XCA layer, and an MLP applied only to the class token. Residual connections are used around XCA and the MLP, mirroring the AFT blocks. By the SEA properties, the patch tokens remain shift-equivariant and the class token is shift-invariant (see Proposition 4). The final prediction is obtained by a linear classifier applied to the class token.

# 4  Experiments

We evaluate our Alias-Free Transformer (AFT) on the ImageNet dataset [12] and compare its accuracy and shift consistency with the baseline XCiT model. We additionally compare our method with the adaptive polyphase sampling (APS) approach [13, 47], which we implement by replacing the strided-convolutions in the PE with stride-1 convolutions followed by APSPool [6], and using the standard class attention blocks for classification (maintaining integer shift-invariance). We use the nano and small XCiT versions with 12 layers and patch-size 16, processing inputs of size $224 \times 224$.

We train all models for 400 epochs, following the XCiT training recipe [16], using PyTorch [41], on a single machine with $8 \times$ NVIDIA RTX A6000. We observe a slight improvement for the AF models when training with a smaller batch size; therefore, we reduce the batch size from 1024 to 512 for the AF versions (See additional details in Appendix D.1).

In Sections 4.1 and 4.2 we evaluate the baseline, APS, and AFT models using cyclic translations and implement $m/n$-fractional translation by translating in $m$ pixels the $n$-upsampled image using sinc-interpolation, as our and the APS models were initially designed under those assumptions. In Section 4.3 we use more "realistic" types of translations, and add additional publicly available ViTs to the comparison.

## 4.1  Accuracy and shift consistency

The results in Table 1 (left) show the classification accuracy and consistency, defined as the percentage of validation samples whose prediction did not change after a random translation. The alias-free

Table 1: **ImageNet accuracy and cyclic shift consistency. Left**: Shift consistency is defined as the percentage of test samples that did not change their prediction following a random translation. The alias-free models have similar accuracy to the baseline models, and much higher consistency in both integer and half-pixel translations. **Right**: Adversarial accuracy is defined as the percentage of correctly classified samples in the worst case at the corresponding grid (Equation (9)). See Appendix B for additional results.

| Model | Test accuracy | Integer shift consist. | Half-pixel shift consist. | Adversarial integer grid | Adversarial half-pixel grid |
|---|---|---|---|---|---|
| XCiT-Nano (Baseline) | 70.4 | 83.7 | 82.0 | 50.9 | 52.9 |
| XCiT-Nano-APS | 68.4 | **100.0** | 87.5 | 68.4 | 62.9 |
| XCiT-Nano-AF (ours) | **70.5** | 99.2 | **98.7** | **69.9** | **69.5** |
| XCiT-Small (Baseline) | **82.0** | 91.4 | 89.8 | 70.9 | 71.3 |
| XCiT-Small-APS | 81.3 | **100.0** | 94.0 | **81.3** | 78.2 |
| XCiT-Small-AF (ours) | 81.8 | 99.5 | **99.4** | 81.3 | **81.1** |

models have similar accuracy to the baseline models, and much higher consistency in both integer and half-pixel translations. The APS models, on the other hand, achieve near-100% consistency under integer translations, as expected. However, they have a more modest improvement in consistency to half-pixel shifts.

## 4.2 Adversarial robustness

To show a practical implication of shift consistency, we ask whether an adversary, free to translate the image within a prespecified grid, can find any shift that flips the label. We define adversarial accuracy as the fraction of images that are classified correctly in the worst-case at this grid. In Table 1 (right), we show results for cyclic integer and half-pixel translations, reporting adversarial accuracy over the following grids

$$T_{\text{integer}} = \left\{ (i,j) \,\middle|\, -6 \leq i,j \leq 6 \right\} \qquad T_{\text{half}} = \left\{ \left(\tfrac{i}{2}, \tfrac{j}{2}\right) \,\middle|\, -6 \leq i,j \leq 6 \right\} \tag{9}$$

The AFT models maintain high accuracy under both integer and half-pixel attacks, having 2% relative accuracy reduction in the nano version and less than 1% reduction in the small model. This is in contrast to the baseline models, with relative accuracy reductions of 25% and 14% in the nano and small models respectively. As expected, the APS models maintain their accuracy under the integer grid adversarial attacks. Their accuracy under half-pixel grid attacks decreases slightly in comparison to the baseline models. The reason for this is that the APS is invariant to any two half-pixel translations, as these differ in exact integer translations. We expect the APS accuracy to decrease more under arbitrary fractional translations, as can be seen in Section 4.3 and [40]. See Appendix B for additional results.

## 4.3 Robustness to realistic shifts

The experiments above use cyclic translations, which may leave unnatural image artifacts in realistic cases (where the input is not a sample of some periodic signal). We therefore test two more realistic perturbations and measure adversarial accuracy exactly as in Section 4.2. See Appendix C for visualizations of the used translations.

- **Crop-shifts.** The image is first center-cropped, then cropped in offsets $(\delta_x, \delta_y)$ with $|\delta_x|, |\delta_y| \leq s$. This mimics a camera translation that moves content out of view instead of wrapping it around.

- **Bilinear fractional shifts.** To simulate sub-pixel motion, we translate the image by $(\delta_x/6, \delta_y/6)$ with $|\delta_x|, |\delta_y| \leq s$, using bilinear interpolation. Here, we leave an edge of one pixel of the original image in each direction to avoid edge artifacts.

We compare the baseline, APS and AFT XCiT-Small models with other publicly available trained models, in similar scale as XCiT-small (26M) (indicates number of parameters): CvT-13 (20M), Swin-T (28M), and ViT-Base (86M). We repeat these experiments with $s$ (max-shift) in the range 0

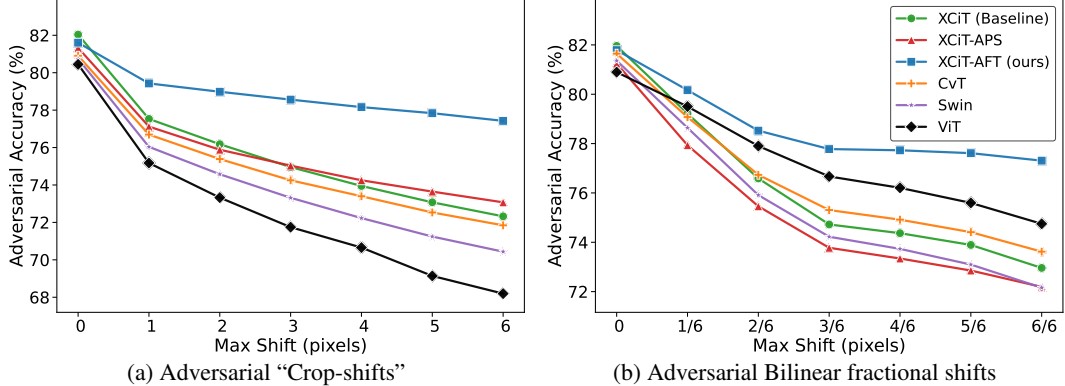

Figure 2: **ImageNet adversarial accuracy under realistic translations.** Adversarial accuracies under (a) "Crop-shifts," simulating camera translations, and (b) "Bilinear fractional shifts," simulating realistic sub-pixel image translations. The Alias-Free Transformer (AFT) consistently outperforms baseline XCiT, APS, and other vision transformer variants (CvT, Swin and ViT), demonstrating superior robustness against realistic translations.

to 6. The results in Figure 2 show our model has improved robustness to both these types of realistic shifts, despite not being specifically designed for them. Among the additional baselines, the vanilla ViT degrades the most under crop-shifts, whereas under bilinear fractional shifts it is surprisingly competitive and in fact better than the hybrid backbones (XCiT, Swin, CvT).

## 4.4 Ablation study

We conduct an ablation study on XCiT-Nano to evaluate the impact of each of the alias-free modifications on the model performance. We train the baseline model on ImageNet with one specific change, and report the results in Table 2. Surprisingly, replacing the layer norm with alias-free layer-norm and replacing class attention with average pooling cause a much larger degradation in accuracy in comparison to the marginal degradation in the final alias-free model. On the other hand, removing the positional encoding leads to a small improvement in accuracy, emphasizing that it may be unnecessary in hybrid architectures.

Table 2: **Ablation study on alias-free components of XCiT-Nano (ImageNet).** Evaluation of isolated alias-free modifications to the baseline model. Alias-free layer normalization (AF-LayerNorm) and replacing class-attention with average pooling (AvgPool) result in notable accuracy degradation individually. Removing positional encoding slightly improves performance. The final combined alias-free model retains near-baseline accuracy.

| Model | Accuracy | Change (%) |
|---|---|---|
| Baseline | 70.4 | – |
| Cyclic convolution | 70.4 | +0.0% |
| AvgPool | 69.1 | −1.8% |
| AF-LayerNorm | 69.6 | −1.1% |
| No positional encoding | 70.7 | +0.4% |
| AF (AvgPool) | 70.4 | +0.0% |
| AF (AF Class Attention) | 70.6 | +0.3% |

Table 3: **Training runtime.** Train time was measured on 8 × NVIDIA RTX A6000 using batch size 1024 for the baseline model and batch size 512 in the APS and AF models, due to memory constraints.

| Model | Train time [hours] |
|---|---|
| XCiT-Small (Baseline) | 69 |
| XCiT-Small-APS | 98 |
| XCiT-Small-AF (ours) | 487 |

# 5   Related work

A few studies have shown a broad effect of aliasing in deep neural networks, e.g., breaking shift-equivariance in convnets [4, 67], inconsistencies in image generation [32, 64, 70], and breaking continuous-discrete equivalence in neural operators [5, 19, 55, 69].

**Shift invariant convnets.**   For a long time, convolutional neural networks have held dominance in vision thanks to their useful inductive biases, including translation invariance. However, previous studies have shown their output can in fact change in a large extent due to small translations [4, 17]. This has led to extensive research to find the root causes and resolve this problem. Azulay and Weiss [4], Zhang [67] have identified shift-invariance breaks as a result of aliasing in downsampling and nonlinear layers. Consequently, other studies suggested solving this problem by plugging a low-pass filter before downsampling [23, 29, 67], and preventing aliasing in nonlinearities by using smooth activation functions [31, 40, 57] and by applying activations after upsampling [32, 40, 60]. Other works suggested maintaining shift-invariance in convnets by downsampling on adaptive grids [6, 46]. Specifically, the adaptive sampling method (APS) [6, 46] has been shown to retain perfect consistency to integer cyclic translations, while the anti-aliasing approach maintains consistency in fractional translations as well. Worth mentioning here are recent works that propose transforming the input into a "canonic" shift-invariant representation [11, 51], which theoretically makes equivariance of the following neural network unnecessary.

**Shift invariant transformers.**   Vision transformers have gained dominance despite not having the convnet priors and being even more sensitive to image translations. Some studies have proposed hierarchical ViT architectures similar to convnets [14, 18, 39, 49, 62], and "hybrid" architectures including convolutional layers directly [61]. Furthermore, some works have drawn parallels between self-attention and dynamic convolutions [1, 9, 27], motivating reinterpretations of the attention mechanism through a convolutional lens. Few studies have taken inspiration from these studies aiming to retain shift-invariance in convnets and implemented their ideas into ViTs. Qian et al. [42] proposes plugging a low-pass filter post self-attention to reduce aliasing, partially improving consistency similar to Zhang [67]. Ding et al. [13], Rojas-Gomez et al. [47] adapt the adaptive sampling method (APS) into ViT layers, certifying consistency to integer cyclic translations. Other studies proposed more general framework for group equivariant attention [48, 63]. Yet, to the best of our knowledge, no other work has dealt with the invariance of ViTs to fractional shifts.

**Linear attention.**   The standard Transformer architecture relies on softmax-based attention [58], characterized by quadratic computational complexity in the number of tokens. To overcome scalability limitations, linear and kernel-based attention mechanisms have been proposed [7, 33], substantially reducing complexity while maintaining performance. For example, linear attention leverages a linear approximation of the softmax kernel, achieving significant efficiency gains [59]. In vision, models like SimA [35] and XCiT [16] utilize simplified normalization schemes to replace the expensive softmax operation, enabling to avoid a direct computation of full attention maps.

# 6   Discussion and limitations

In this paper, we propose a shift-invariant alias-free vision transformer by introducing a class of shift-equivariant attention operations. We show that the AFT maintains competitive accuracy and superior robustness to fractional shifts, compared to other ViTs. We next discuss a few of our model limitations.

**Polynomial activation function**   Michaeli et al. [40] propose replacing nonlinear activation functions, such as GELU, with polynomial approximations. This mathematically ensures the overall activation layer (including upsampling) is shift-equivariant w.r.t. continuous domain, namely perfectly consistent to fractional shifts. In our experiments, we observe that this leads to a significant reduction in performance, which is caused specifically due to the activation replacement in the patch-embedding stage (see Appendix B). On the other hand, we observe that the filtered activation function using GELU leads to a rather small reduction in consistency. Notably, the certified consistency is limited to cyclic shifts and fractional shifts performed by sinc-interpolation, both induce artifacts that do not

appear in natural images. We find that similar to the AFC, our model also has significant improvement in robustness to realistic translations despite the imperfect consistency to cyclic shifts.

**Runtime performance**   The alias-free modifications we perform in our model to attain shift invariance, despite not requiring any additional parameter, cause a substantial runtime increase, as shown in Table 3. This is mainly due to the downsampling and upsampling, which are implemented in the Fourier domain using FFT, similar to other works [22, 23, 40, 43, 70], seemingly underoptimized for GPU as of today [20, 53].

## Acknowledgments and Disclosure of Funding

The research of DS was funded by the European Union (ERC, A-B-C-Deep, 101039436). Views and opinions expressed are however those of the author only and do not necessarily reflect those of the European Union or the European Research Council Executive Agency (ERCEA). Neither the European Union nor the granting authority can be held responsible for them. DS also acknowledges the support of the Schmidt Career Advancement Chair in AI.

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

# A  Full proofs

**Proposition 2.**  $f\left(K^\top V\right)$ is shift-invariant.

*Proof.* The problem can be simplified by considering an arbitrary entry in $K^\top V$, since

$$\left(K^\top V\right)_{i,j} = K_i^\top V_j \,,$$

where $K_i$, $V_j$ are the $i$-th and $j$-th columns of $K$ and $V$, representing 1-dimensional signals.

By Parseval theorem, this inner product between signals equals to their inner product in Fourier domain:

$$K_i^\top V_j = \frac{1}{2\pi} \int_{-\pi}^{\pi} \hat{K}_i\left(\omega\right) \hat{V}_j^{\ *}\left(\omega\right) d\omega \,. \tag{10}$$

Denote $K^\tau$ and $V^\tau$ the queries and values of a $\tau$-shifted input signal w.r.t. the input signal of $K$ and $V$. Following Proposition 1, a translation by $\tau$ of the input $x$ yields a translation by $\tau$ of both $K$ and $V$. The Fourier transform of the translated signals differs by a phase which cancels out in the inner product, thus we get the same product:

$$K_i^{\tau\top} V_j^\tau = \frac{1}{2\pi} \int_{-\pi}^{\pi} \hat{K}_i^\tau\left(\omega\right) \hat{V}_j^{\tau\ *}\left(\omega\right) d\omega \tag{11}$$

$$= \frac{1}{2\pi} \int_{-\pi}^{\pi} \hat{K}_i\left(\omega\right) e^{j\omega\tau} \left(\hat{V}_j\left(\omega\right) e^{j\omega\tau}\right)^* d\omega \tag{12}$$

$$= \frac{1}{2\pi} \int_{-\pi}^{\pi} \hat{K}_i\left(\omega\right) \hat{V}_j^{\ *}\left(\omega\right) d\omega \tag{13}$$

$$= K_i^\top V_j \,. \tag{14}$$

$\square$

**Class Attention.**  Below we formalize the statement regarding shift-invariance of the Class Attention layer in Section 2.3.

**Proposition 4.** *Let $Q = XW_q$, $K = XW_k$ and $V = XW_v$ be the query, key and value matrices of a patch sequence $X \in \mathbb{R}^{N \times D}$. Append the sequence with a class token,*

$$\tilde{Q} = \left[Q^\top, q_{\text{cls}}\right]^\top, \qquad \tilde{K} = \left[K^\top, k_{\text{cls}}\right]^\top, \qquad \tilde{V} = \left[V^\top, v_{\text{cls}}\right]^\top, \tag{15}$$

*with learnable vectors $q_{\text{cls}}, k_{\text{cls}}, v_{\text{cls}} \in \mathbb{R}^D$. Our CA layer applies the SEA update*

$$\tilde{V}' = \text{SEA}\left(\tilde{Q}, \tilde{K}, \tilde{V}\right) = \tilde{Q} f\left(\tilde{K}^\top \tilde{V}\right) \in \mathbb{R}^{(N+1) \times D}, \tag{16}$$

*where $f \colon \mathbb{R}^{(D \times D)} \to \mathbb{R}^{D \times D}$ is any matrix function. Denote the output by $\tilde{V}' = \left[V'^\top, v'_{\text{cls}}\right]^\top$ with $V' \in \mathbb{R}^{N \times D}$. Then*

1. ***Patch equivariance:** $V'$ is shift-equivariant.*

2. ***Class invariance:** $v'_{\text{cls}}$ is shift-invariant.*

*Proof.* Let $Q^\tau$ $K^\tau$ and $V^\tau$ be the keys and values obtained from the $\tau$-translated input $X$, and define

$$\tilde{Q}^\tau = \left[\left(Q^\tau\right)^\top, q_{\text{cls}}\right]^\top, \quad \tilde{K}^\tau = \left[\left(K^\tau\right)^\top, k_{\text{cls}}\right]^\top, \quad \tilde{V}^\tau = \left[\left(V^\tau\right)^\top, v_{\text{cls}}\right]^\top. \tag{17}$$

**Shift-invariance of the attention weights.** Since the translation $\tau$ acts only on patch tokens, we get

$$\tilde{K}^{\tau\top} \tilde{V}^\tau = \left(K^\tau\right)^\top V^\tau + k_{\text{cls}} v_{\text{cls}}^\top = K^\top V + k_{\text{cls}} v_{\text{cls}}^\top = \tilde{K}^\top \tilde{V}, \tag{18}$$

where the middle equality uses Proposition 2. The rank-one class term $k_{\text{cls}} v_{\text{cls}}^\top$ is constant (independent of the input translation), hence $f(\tilde{K}^\top \tilde{V})$ is shift-invariant.

It holds that

$$\tilde{V}' = \tilde{Q} f\left(\tilde{K}^\top \tilde{V}\right) = \left[ Q^\top f\left(\tilde{K}^\top \tilde{V}\right), \, q_{\text{cls}}^\top f\left(\tilde{K}^\top \tilde{V}\right) \right]^\top \tag{19}$$

**Patch equivariance.** From Equation (19), the patch tokens post CA are $V' = Q^\top f\left(\tilde{K}^\top \tilde{V}\right)$, where $f\left(\tilde{K}^\top \tilde{V}\right)$ is shift-invariant, therefore $V'$ is shift-equivariant similar to Proposition 3.

**Class invariance.** From Equation (19), the class token post CA is $q_{\text{cls}}^\top f\left(\tilde{K}^\top \tilde{V}\right)$, which is shift-invariant. □

# B Additional results

## B.1 Additional datasets

We evaluate all models from section 4.1 on three additional classification benchmarks — CIFAR-10, CIFAR-100 [37], and Stanford Cars [36] — under two protocols: (i) fine-tuning ImageNet-pretrained checkpoints and (ii) training from scratch. In both protocols, we train each model using the ImageNet recipe of Table 8 with 1,000 epochs, where in the fine-tuning protocol, we initialize the model weights using the checkpoints from section 4.1. We report top-1 accuracy together with cyclic shift consistency for integer and half-pixel translations, defined exactly as in Section 4.1.

Across all datasets, AFT maintains near-perfect shift-equivariance (above 99% consistency in integer and half-pixel shifts). Additionally, when fine-tuned, the XCiT-Small-AF model is slightly but consistently more accurate than the baseline on all three datasets, suggesting that the shift-invariance prior can benefit larger transformers when adapting to small datasets.

Table 4: **CIFAR and Stanford Cars (fine-tuning): accuracy and cyclic shift consistency.** We fine-tune ImageNet-pretrained checkpoints on CIFAR-10/100 and Stanford Cars. Metrics are top-1 test accuracy and consistency to integer and half-pixel cyclic translations (as in Section 4.1).

| | **CIFAR-10** | | | **CIFAR-100** | | | **Stanford Cars** | | |
|---|---|---|---|---|---|---|---|---|---|
| **Model** | Test accuracy | Integer shift consist. | Half-pixel shift consist. | Test accuracy | Integer shift consist. | Half-pixel shift consist. | Test accuracy | Integer shift consist. | Half-pixel shift consist. |
| XCiT-Nano (Baseline) | **98.0** | 98.0 | 98.1 | **84.3** | 89.4 | 89.9 | **92.3** | 95.2 | 95.1 |
| XCiT-Nano-APS | 97.7 | **100.0** | 99.4 | 84.1 | **100.0** | 96.9 | 92.2 | **100.0** | 97.0 |
| XCiT-Nano-AF (ours) | 97.7 | 99.9 | **99.9** | **84.3** | 99.6 | **99.4** | 92.1 | 99.9 | **99.8** |
| XCiT-Small (Baseline) | 98.2 | 98.4 | 98.5 | 85.6 | 87.4 | 89.6 | 92.6 | 95.4 | 96.3 |
| XCiT-Small-APS | 98.3 | **100.0** | 99.5 | 85.4 | **100.0** | 96.2 | 92.2 | **100.0** | 97.2 |
| XCiT-Small-AF (ours) | **98.4** | 99.9 | **99.9** | **85.8** | 99.6 | **99.5** | **93.0** | 99.9 | **99.9** |

Table 5: **CIFAR and Stanford-Cars (from scratch): accuracy and cyclic shift consistency.** Models are trained from scratch on each dataset using the ImageNet training setup with 1,000 epochs; metrics as in Section 4.1.

| | **CIFAR-10** | | | **CIFAR-100** | | | **Stanford Cars** | | |
|---|---|---|---|---|---|---|---|---|---|
| **Model** | Test accuracy | Integer shift consist. | Half-pixel shift consist. | Test accuracy | Integer shift consist. | Half-pixel shift consist. | Test accuracy | Integer shift consist. | Half-pixel shift consist. |
| XCiT-Nano (Baseline) | **97.2** | 98.0 | 98.0 | **82.4** | 90.3 | 90.2 | **86.5** | 91.6 | 91.7 |
| XCiT-Nano-APS | 97.2 | **100.0** | 99.2 | 82.3 | **100.0** | 97.1 | 84.1 | **100.0** | 93.3 |
| XCiT-Nano-AF (ours) | 96.5 | 99.9 | **99.9** | 81.3 | 99.5 | **99.4** | 85.3 | 99.7 | **99.6** |
| XCiT-Small (Baseline) | **98.3** | 98.7 | 98.7 | **85.3** | 91.3 | 91.3 | 89.6 | 95.1 | 94.5 |
| XCiT-Small-APS | 98.0 | **100.0** | 99.4 | 85.1 | **100.0** | 95.7 | **90.5** | **100.0** | 96.5 |
| XCiT-Small-AF (ours) | 97.6 | 99.9 | **99.9** | 83.4 | 99.6 | **99.6** | 88.5 | 99.8 | **99.8** |

## B.2 Global average pooling vs AF Class Attention

In section 2 we propose two mechanisms to get a shift-invariant global representation out of the AFT — a global average pooling over the embedding dimension and an alias-free class attention that leverages SEA to maintain a shift-invariant class token. We compare the final *AF class attention* (AFCA) head with a *global average pooling* (AvgPool) head within the AFT architecture. As shown in Table 6, both AFCA and AvgPool maintain near-perfect shift consistency. AFCA demonstrates consistent improvement over AvgPool in top-1 accuracy, most visible in the Small variant.

Table 6: **AF class attention vs. global average pooling (AFT).** Top-1 accuracy and cyclic shift consistency (integer and half-pixel) on ImageNet. AvgPool and AFCA retain near-perfect equivariance, while AFCA provides a consistent accuracy gain, most notably for the Small variant.

| Model | Test accuracy | Integer shift consist. | Half-pixel shift consist. |
|---|---|---|---|
| XCiT-Nano-AF (AvgPool) | 70.35 | 99.0 | 98.6 |
| XCiT-Nano-AF (AF-CA) | 70.48 | 99.2 | 99.4 |
| XCiT-Small-AF (AvgPool) | 80.70 | 99.5 | 99.4 |
| XCiT-Small-AF (AF-CA) | 81.81 | 99.5 | 99.4 |

### B.3   Polynomial vs GELU comparison

Similar to the Alias-Free ConvNet (AFC) of Michaeli et al. [40], certified shift-invariance in the Alias-Free Transformer (AFT) can be achieved by replacing the filtered GELU activations with polynomial approximations. We therefore train an alias-free XCiT-Nano variant whose activations are polynomials with learnable coefficients per embedding channel, following Michaeli et al. [40]. We use polynomials of degree 2 in the AFT blocks and degree 3 in the patch-embedding stage (PE), which remains alias-free thanks to the downsampling layers following the activations in the PE. The results in Table 7 show that the full polynomial model ("Poly") has near-100% shift consistency, with a small gap that can be attributed to numerical errors, similar to the case in the APS model (see Table 1). However, we observe that unlike in the AFC, polynomial activations lead to a significant reduction in accuracy.

Interestingly, when the four GELU activations in the PE are retained and only the block activations are replaced ("GELU (PE), Poly (Blocks)"), most of the lost accuracy is recovered. This may indicate that polynomial activations limit the representational capacity of the convolutional PE which is much shallower than the convnet tested in AFC.

Table 7: **Effect of polynomial activations on ImageNet performance and shift consistency.** Top-1 accuracy and consistency (%) of XCiT-Nano with the standard filtered GELU, full polynomial replacement (Poly), and a hybrid that keeps GELU in the patch-embedding (PE)

| Model | Test accuracy | Integer shift consist. | Half-pixel shift consist. |
|---|---|---|---|
| GELU | 70.4 | 98.8 | 98.4 |
| Poly | 65.8 | 99.7 | 99.6 |
| GELU (PE), Poly (Blocks) | 68.5 | 99.4 | 98.7 |

## C   Translation visualization

We provide visual examples of the translations described in the paper in Figures 3 and 4.

## D   Experiments details

As mentioned in Section 4, we used the same training settings as in XCiT [16], except for lowering the batch size for the AF model. We report the used training hyperparameters in Table 8.

### D.1   Batch size choice

Aiming to avoid expensive hyperparameter tuning, we used the XCiT original recipe [16]. However, in our early experimentations, we observed fluctuations in the training curves of the AF and APS models, which were alleviated by reducing the batch size. To decide fair batch sizes, we trained the *Nano* variants of Baseline, APS, and AF with batch sizes 512 and 1024 and picked, for each method, the configuration that performed best. The Baseline favored 1024 (top-1 70.4 vs. 70.1 at 512), while

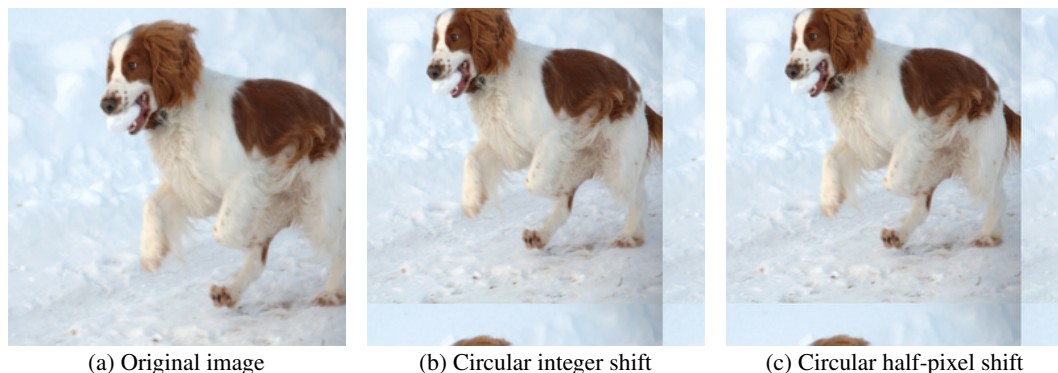

|  (a) Original image  |  (b) Circular integer shift  |  (c) Circular half-pixel shift  |

Figure 3: **Visualization of cyclic shifts.** (a) Original ImageNet [12] validation-set image. (b) Circular shift of 16 pixels in horizontal and vertical axes. (c) Circular shift of 16.5 pixels in horizontal and vertical axes. The original image is upsampled by a factor 2, circularly shifted by 33 pixels, and downsampled by factor 2.

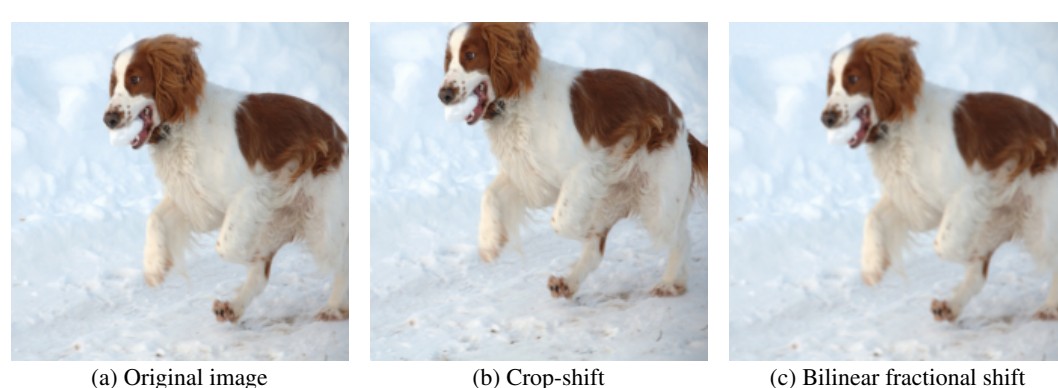

|  (a) Original image  |  (b) Crop-shift  |  (c) Bilinear fractional shift  |

Figure 4: **Visualization of realistic shifts.** (a) Original ImageNet [12] validation-set image — $224 \times 224$ center crop of the original $256 \times 256$ image. (b) Crop-shift of the original image of 16 pixels in the horizontal and vertical axes. The cropped area is shifted by 16 pixels with respect to the cropped area in the original image. (c) Bilinear fractional shift of 0.5 pixels in horizontal and vertical axes. We use a $226 \times 226$ center crop of the original $256 \times 256$ image and simulate a fractional-pixel shift using a grid-sample with a fractional offset.

Table 8: **Hyperparameters.** Unless stated otherwise, the same settings apply to all models.

| Category | Parameter | Value |
|---|---|---|
| Optimizer | Optimizer | AdamW |
| | $(\beta_1, \beta_2)$ | (0.9, 0.999) |
| | Weight decay | 0.05 |
| Learning rate scheduling | Base LR | $1 \times 10^{-3}$ (Baseline), $5 \times 10^{-4}$ (AF, APS) |
| | Warm-up epochs | 5 |
| | LR decay | Cosine |
| | Min LR | $1 \times 10^{-5}$ |
| Data | Resolution | $224 \times 224$ |
| | Batch size | 1024 (Baseline), 512 (AF, APS) |
| Regularization | Layer scale ($\epsilon$ init) | 1.0 |
| | Stochastic depth | 0.0 (Nano), 0.05 (Small) |

APS and AF favored 512 (APS: 68.7 vs. 67.2 at 1024; AF: 70.4 vs. 70.1 at 1024). We therefore used batch size 1024 for the Baseline and 512 for APS and AF throughout the paper. Note that the figures

above for the AF and APS models are with an AvgPool head. We applied the same choice to the Nano and Small variants with CA and AFCA heads without further tuning.

