# OpenReview forum: "Alias-Free ViT: Fractional Shift Invariance via Linear Attention"
_NeurIPS.cc/2025/Conference — NeurIPS 2025 poster_

### Official Review · Reviewer_GB1B · 2025-06-20

**Clarity:** 3
**Significance:** 2
**Originality:** 3
**Rating:** 4
**Confidence:** 4

**Summary:**

The authors propose a modification of the vision transformer architecture based on XCiT (that uses linear attention) to ensure invariance to fractional shifts. The authors theoretically establish shift invariance by stepping through each component of the architecture and empirically validate the method on ImageNet classification. The authors compare against XCiT using the small and nano architectures as well as a shift invariant baseline method APS. The authors also compare the training time required for each method.

**Questions:**

- can you include additional figures for what the natural shifts used in Figure 2 are? It would be great to show for example an original image and transformed variants in contrast to the fractional / integer shifts that the authors present as more artificial. This would help illustrate to readers the benefit of the method as well.

**Ethical Concerns:**

["NO or VERY MINOR ethics concerns only"]

**Final Justification:**

This work proposes a modification to the popular vision architecture to tackle fractional shifts—while a nice contribution, the limited scope of the problem and limited empirical verification in terms of hyperparameters are the primary justifications for the rating.

**Limitations:**

yes

**Quality:**

3

**Strengths And Weaknesses:**

* It's nice to see the authors building on prior work and using more efficient linear attention as the foundation for their approach.
* I also appreciate the clarity of the exposition overall, particularly in establishing theoretical shift invariance for the proposed architecture.
* Importance of fractional shifts. Figure 2: shows the importance of the issue of fractional intelligence in translations that are likely to come up in real settings. I recommend the authors highlight this finding to illustrate the importance of fractional shifts as a research problem.

* It would be good to include a vanilla ViT as an additional baseline or reference to illustrate the gains from linear attention and lack of invariance to translation for the dominant ViT architecture.
* The proposed method comes with several tradeoffs. First, compared to APS the method achieves lower test accuracy for both the nano and small XCiT architectures. Second, the method achieves slightly lower integer shift consistency, particularly for the nano architecture (Table 1). Third, and perhaps most important is that the method comes with a much higher training cost (509 hours versus 303 for APS and only 77 for the baseline XCiT small model). This is a significant cost that must be clearly highlighted to make the trade off clear to readers, particularly in the abstract, which seems to ignore these important tradeoffs.
    * Specifically the abstract read "Our model maintains competitive performance in image classification and outperforms similar‑sized models in terms of robustness to adversarial translations" while omitting discussion of these tradeoffs. I recommend including a more detailed discussion of these tradeoffs in the introduction as well.
* One other suggestion to bolster the experiments is to consider also adding object detection and segmentation experiments to further validate the proposed architecture, as was done in prior work such as XCiT.
* I acknowledge resource constraints may not make this possible and believe the current experiments on their own are certainly interesting, but I would be curious what happens when the both the model and training data is scaled (to say ImageNet-22k or larger).  The original ViT paper for example demonstrated the performance of the transformer architecture in vision can change significant with scale (and that ViT is generally data hungry). If the authors are resource constrained, I would recommend at least noting the scale of the experiments (model sizes and training data size) as limitations.

---

> ### Author Rebuttal · Authors · 2025-07-30
>
> We thank the reviewer for their endorsement and constructive feedback. We are happy the reviewer found our work clearly explained and well-motivated.
> In the following, we address the questions that have been raised. We will incorporate these clarifications and additional results into the paper (or its appendix).
>
> >   It would be good to include a vanilla ViT as an additional baseline or reference to illustrate the gains from linear attention and lack of invariance to translation for the dominant ViT architecture.
>
> We thank the reviewer for this suggestion. We find that vanilla ViT is indeed less robust to crop shifts than the hybrid models, and our AFT in particular. For example, ViT-Base adversarial accuracy for crop shifts with max-shift 3 (see Figure 2a) declines from 80.4 to 71.7 vs. 80.0 to 76.4 in our model. We will include complete results in the revision of the paper.
>
> >   The proposed method comes with several tradeoffs. First, compared to APS the method achieves lower test accuracy for both the nano and small XCiT architectures. Second, the method achieves slightly lower integer shift consistency, particularly for the nano architecture (Table 1).
>
> *Table 1 had an issue which gave APS an unfair advantage, but we already corrected this issue in Section B of the SM (Table 4)*.
> Specifically, upon the main paper submission, we detected an error in our implementation of the APS model (in the results presented in Table 1 and Figure 2), leading to training with an effective patch size of 8 instead of 16, as was used in the Baseline and AF models. This led to better accuracy for the APS model and an unfair comparison vs. the other methods.
>
> The corrected results in Table 4 (in the SM) show that the APS model has lower accuracy than the AF model (APS 68.7 vs. AF 70.6 in the Nano versions and APS 80.1 vs. AF 80.7 in the Small versions). Consequently, the APS and AF models have similar adversarial accuracy under the integer grid (APS 68.7 vs. AF 68.9 in the Nano versions and APS 80.1 vs. AF 80.0 in the Small versions). Notably, the AF model has better adversarial accuracy under the half-pixel grid, with a larger margin than reported in Table 1 (APS 62.9 vs. AF 69.2 in the Nano versions and APS 76.0 vs. AF 79.9 in the Small versions).
>
> > Third, and perhaps most important, is that the method comes with a much higher training cost (509 hours versus 303 for APS and only 77 for the baseline XCiT small model). This is a significant cost that must be clearly highlighted to make the trade-off clear to readers, particularly in the abstract, which seems to ignore these important trade-offs.
>
> Indeed, the proposed model currently comes with a high additional computational cost. As mentioned in the discussion, we believe this issue may be alleviated by carefully optimizing components that are currently not commonly used within neural architectures. For example, [A] reports a custom CUDA kernel for their alias-free activation function, which uses an approximation of sinc-interpolation upsampling, yielding a 10× training speedup. Nevertheless, we will make sure this trade-off is clearly noted.
>
> >   One other suggestion to bolster the experiments is to consider also adding object detection and segmentation experiments to further validate the proposed architecture, as was done in prior work such as XCiT.
>
> We thank the reviewer for the suggestion. We also find it intriguing to implement translation-robust detection and segmentation models using the anti-aliasing approach. XCiT is indeed used for detection and segmentation as a backbone to Mask R-CNN. However, this includes additional modules that are not trivial to convert to be fractional-shift equivariant under our framework (for example, RoI Align, which uses bilinear interpolation). Therefore, we leave this as an interesting direction for future work and focus this paper on classification tasks.
>
> >   I acknowledge resource constraints may not make this possible and believe the current experiments on their own are certainly interesting, but I would be curious what happens when the both the model and training data is scaled (to say ImageNet-22k or larger). The original ViT paper for example demonstrated the performance of the transformer architecture in vision can change significant with scale (and that ViT is generally data hungry). If the authors are resource constrained, I would recommend at least noting the scale of the experiments (model sizes and training data size) as limitations.
>
> Unfortunately, we were unable to scale the size of the experiments due to limited resources. We will mention this in the limitations section of the camera-ready version.
>
> > Can you include additional figures for what the natural shifts used in Figure 2 are? It would be great to show for example an original image and transformed variants in contrast to the fractional / integer shifts that the authors present as more artificial. This would help illustrate to readers the benefit of the method as well.
>
> We thank the reviewer for this suggestion. Unfortunately, we are unable to attach figures in the rebuttal, but we will include a visual example in the revision. We kindly refer the reviewer to visual examples in [B], where Figure 6 demonstrates circular shift and crop shift, and Figure 5 demonstrates bilinear fractional shift. Note that, unlike in Figure 5, we interpolated the translated images while leaving an edge of one pixel instead of padding with zeros, to avoid any edge artifacts
>
> ---
>
> [A] Alias-Free Generative Adversarial Networks, NeurIPS, 2021.
>
> [B] Alias-free convnets: Fractional shift invariance via polynomial activations. arXiv, 2023.

---

> > ### Comment · Reviewer_GB1B · 2025-08-01
> >
> > Thank you for the clarifications and responses to my questions. Given, your comment regarding the effect of batch size, I'm concerned to what extent the overall results are robust to hyperparameter choices such as batch size. This concern is also highlighted by other reviewers (and authors response "We refrained from performing full hyperparameter tuning due to limited resources" so far does not give me much comfort). Can you comment on the choice hyperparameter selection and robustness of the results to choices such as batch size?

---

> > > ### Author Response · Authors · 2025-08-03
> > >
> > > We thank the reviewer for allowing us to clarify this. Our policy was to keep all hyperparameters identical to the XCiT baseline [A] across models, since our method is an architectural modification of XCiT. This is consistent with APS, which was also trained in [B] with the same baseline hyperparameters for each architecture.
> > >
> > > The only deviation was motivated by training stability: for the AF model, we observed larger loss fluctuations at batch size 1024, which were alleviated by reducing the batch size to 512 and scaling the learning rate proportionally (as in [A]). Ultimately, this has led to a slight improvement in accuracy. We did the same procedure for the other nano models (choosing the best batch size for the Baseline, APS and AF individually, to maintain fairness) and consequently reduced the APS batch size as well. We then used the same configuration for the small versions of each model (i.e., 1024 for the baseline and 512 for APS and AF).
> > >
> > > To be transparent about sensitivity to this single change, we reported in response to reviewer he7x results for both batch sizes on the Nano models:
> > > * XCiT-Nano (Baseline): 70.1 @ 512 vs 70.6 @ 1024
> > > * XCiT-Nano APS: 68.7 @ 512 vs 67.2 @ 1024
> > > * XCiT-Nano-AF (Ours): 70.4 @ 512 vs 70.1 @ 1024
> > >
> > > These two configurations suggest that accuracy variations are modest for all methods under the batch-size change with matched LR scaling. We did not conduct broader sweeps beyond choosing between these two batch-sizes (and their corresponding LR)/LR pair.
> > >
> > > To address your concern in the paper, we will:
> > > * Explicitly document the batch-size/LR choice per model and the stability motivation in the main text/appendix.
> > > * Release the training code and exact training configurations.
> > >
> > > Overall, while we avoided full hyperparameter tuning, the evidence we have indicates that our key claims are not artifacts of a narrow hyperparameter setting. Arguably, the fact that we could avoid significant hyperparameter tuning, despite the significant modifications we applied to the baseline model, can also be seen as a strength of our method (since further HP tuning may have improved the AFs results, but we did not need that).
> > >
> > > ---
> > >
> > > [A] XCiT: Cross-Covariance Image Transformers, NeurIPS 2021.
> > >
> > > [B] Making Vision Transformers Truly Shift-Equivariant, CVPR 2024.

---

### Official Review · Reviewer_N1LY · 2025-07-02

**Clarity:** 3
**Significance:** 2
**Originality:** 2
**Rating:** 4
**Confidence:** 4

**Summary:**

The authors pointed out that ViTs are not translation-invariant and are more sensitive to minor image translations than convnets. To overcome that, the authors proposed the  Alias-Free ViT, which utilizes several shift-invariant models, including alias-free
down-sampling and non-linearities, linear cross-covariance attention. The results show that Alias-Free ViT achieves better robustness to adversarial translations.

**Questions:**

Please see the weakness part.

**Ethical Concerns:**

["NO or VERY MINOR ethics concerns only"]

**Final Justification:**

Thanks for the detailed response. It solved my concerns on the novelty and evaluation. Therefore, I decide to improve the score to Borderline accept.

**Limitations:**

Yes.

**Paper Formatting Concerns:**

null

**Quality:**

2

**Strengths And Weaknesses:**

Pros:

- This paper is well-motivated. To be specific, Alias-Free ViT is proposed to overcome the sensitivity of ViT to minor image translations, which is a reasonable starting point.

- The authors give module-by-module analysis in order to get an alias-free ViT and implement an Alias-Free ViT based on XCiT architecture.

Cons:

- Based on [1], this work seems to be incremental. Even though this paper focuses on transformer architecture, there are many insights originating from [1], e.g., MLP, layer normalization.

- The experimental results are not enough to illustrate the superiority of Alias-Free ViT. For example, the authors only evaluate XCiT, while there are many ViT variants. Also, the baselines (only APS) are not sufficient to make the result convincing.

- As shown in Table 1, the Alias-Free version of XCiT cannot beat XCiT-AFS in 3 aspects out of 5.

[1] Alias-free convnets: Fractional shift invariance via polynomial activations. CVPR, 2023.

---

> ### Author Rebuttal · Authors · 2025-07-30
>
> We thank the reviewer for their constructive feedback. We are happy the reviewer found our work well-motivated.
> In the following, we address the concerns that have been raised. We will incorporate these clarifications into the paper (or its appendix).
>
> >   Based on [A], this work seems to be incremental. Even though this paper focuses on transformer architecture, there are many insights originating from [A], e.g., MLP, layer normalization.
>
> Indeed, many shared components between ViTs and convnets were already “solved.” Therefore, this work is focused on the attention mechanism, as this is the main difference between the two architectures. We believe this work may also have a broader impact, as the attention mechanism is commonly used in many architectures. For example, [B] used existing alias-free components to improve shift equivariance, yet the attention mechanism was overlooked (i.e., it was not modified and, therefore, was not alias-free).
>
> Moreover, we report in the paper additional methodologies and findings that did not originate in [A]. For example, we found that we need to remove the positional encoding and class attention, and to use GeLU instead of polynomial activations in the patchifier. Also, we found that the AF-LayerNorm performs better in the AFT compared to the AFC (in contrast, in AFC it was the main cause of performance degradation). Combining all these small modifications and insights was critical to achieving good performance, and their discovery required careful and extensive experimentation.
>
> >  The experimental results are not enough to illustrate the superiority of Alias-Free ViT. For example, the authors only evaluate XCiT, while there are many ViT variants. Also, the baselines (only APS) are not sufficient to make the result convincing.
>
> We assume this comment relates to Table 1, where we compare three variants (Baseline, APS, and AF (ours)) of XCiT, as this is our baseline model. The main idea of this table is to show that: (1) our model maintains classification accuracy, and (2) it is almost perfectly robust to circular shifts, as theoretically claimed. We compare our model to other ViT variants (Swin and CvT) in Figure 2, where we evaluate “standard shifts” that better represent practical cases. Note that common models usually perform worse on circular shifts than on standard shifts, as shown in [C].
>
> We are unaware of any other relevant baselines (i.e., techniques to improve vision transformers’ translation robustness) with public implementation, other than [C] and [D], represented by APS. We would be happy to add additional baselines to the camera-ready version in case we have missed any.
>
> > As shown in Table 1, the Alias-Free version of XCiT cannot beat XCiT-AFS in 3 aspects out of 5.
>
> *Table 1 had an issue which gave APS an unfair advantage, but we already corrected this issue in Section B of the SM (Table 4)*.
> Specifically, upon the main paper submission, we detected an error in our implementation of the APS model (in the results presented in Table 1 and Figure 2), leading to training with an effective patch size of 8 instead of 16, as was used in the Baseline and AF models. This led to better accuracy for the APS model and an unfair comparison vs. the other methods.
>
> The corrected results in Table 4 (in the SM) show that the APS model has lower accuracy than the AF model (APS 68.7 vs. AF 70.6 in the Nano versions and APS 80.1 vs. AF 80.7 in the Small versions). Consequently, the APS and AF models have similar adversarial accuracy under the integer grid (APS 68.7 vs. AF 68.9 in the Nano versions and APS 80.1 vs. AF 80.0 in the Small versions). Notably, the AF model has better adversarial accuracy under the half-pixel grid, with a larger margin than reported in Table 1 (APS 62.9 vs. AF 69.2 in the Nano versions and APS 76.0 vs. AF 79.9 in the Small versions).
>
>   ---
> [A] Alias-free convnets: Fractional shift invariance via polynomial activations. CVPR, 2023.
>
> [B] Alias-Free Latent Diffusion Models: Improving Fractional Shift Equivariance of Diffusion Latent Space. CVPR, 2025
>
> [C] Making Vision Transformers Truly Shift-Equivariant. CVPR 2024
>
> [D] Reviving Shift Equivariance in Vision Transformers, arXiv 2023

---

> > ### Comment · Reviewer_N1LY · 2025-08-05
> >
> > Thanks for the detailed response. It solved my concerns on the novelty and evaluation. Therefore, I decide to improve the score to Borderline Accept.

---

### Official Review · Reviewer_jU3t · 2025-07-03

**Clarity:** 3
**Significance:** 4
**Originality:** 3
**Rating:** 5
**Confidence:** 3

**Summary:**

The authors, through a series of architectural modifications, present an alias-free vision transformer (ViT) that is invariant to both integer and fractional translations. In particular, they remove positional embeddings, use alias-free convolutional embeddings, replace self-attention with cross-covariance attention, and implement alias-free activation and layer normalization. They then experimentally validate superior performance to existing models on fractional shifts.

**Questions:**

No questions.

**Ethical Concerns:**

["NO or VERY MINOR ethics concerns only"]

**Final Justification:**

The authors present a vision transformer that surpasses previous work in achieving invariance to fractional shifts, rather than just integer shifts. While their approach largely involves recombining many known methods, I am confident that the result is greater than the sum of its parts. I am also less concerned than other reviewers about any potential hyperparameter tuning that may have taken place. Thus, I stand by my original "Accept" rating.

**Limitations:**

Limitations are discussed in Section 6.

**Quality:**

3

**Strengths And Weaknesses:**

**Strengths**
* The presented work, to the best of my knowledge, seems to be the first to focus on invariance of ViTs with respect to fractional shifts.
* An ablation study was performed analyzing the impact of the individual changes on the overall model performance.

**Weaknesses**
* The proposed model can only outperform existing models when evaluated on fractional translations. However, based on the results reported in Tab. 1, the difference decreases substantially for the larger model, raising the question of relevance on even larger ViTs.
* Fractional shifts are evaluated synthetically, e.g., using bilinear interpolation, which may not be representative of fractional shifts observed in real-world distributions. However, it is noted that this is a general problem within this domain and not particular to this work.

---

> ### Author Rebuttal · Authors · 2025-07-30
>
> We thank the reviewer for their endorsement and constructive feedback. We are happy that the reviewer found our work novel.
> In the following, we address the concerns that have been raised. We will incorporate these clarifications and additional results into the paper (or its appendix).
> > The proposed model can only outperform existing models when evaluated on fractional translations. However, based on the results reported in Table 1, the difference decreases substantially for the larger model, raising the question of relevance on even larger ViTs.
>
> Unfortunately, we were unable to evaluate our method on larger models due to limited resources. However, in the following, we evaluate the shift consistency of larger variants of the baseline model (XCiT [A]), as in the third and fourth columns of Table 1 in the paper. Seemingly, the improvement in shift consistency by increasing model size is limited and never reaches the consistency levels of our AF model (>98).
>
> | Model 	          | Params | Integer shift consistency  | Half-pixel consistency |
> |---------------------|--------|----------------------------|------------------------|
> |XCiT-Nano 12 (Tab. 1) | 3M     | 83.2 						| 81.4					|
> |XCiT-Tiny 12         | 7M     | 87.5 						| 86.5					|
> |XCiT-Small 12 (Tab. 1)| 26M    | 90.3 						| 88.8					|
> |XCiT-Small 24        | 48M    | 92.2 						| 90.9					|
> |XCiT-Medium 24       | 84M    | 92.6 						| 91.2					|
> |XCiT-Large 24        | 189M   | 93.0 						| 91.4					|
>
>
> > Fractional shifts are evaluated synthetically, e.g., using bilinear interpolation, which may not be representative of fractional shifts observed in real-world distributions. However, it is noted that this is a general problem within this domain and not particular to this work.
>
> Indeed, “natural translations” (e.g., due to physical camera movements) are hard to simulate precisely given a pixel representation of an image.
>
> ---
> [A] XCiT: Cross-Covariance Image Transformers, NeurIPS 2021.

---

> > ### Comment · Reviewer_jU3t · 2025-08-01
> >
> > I appreciate the additional consistency metrics provided for larger XCiT models that address one of my key concerns and stand by my original rating.

---

### Official Review · Reviewer_he7X · 2025-07-03

**Clarity:** 4
**Significance:** 2
**Originality:** 3
**Rating:** 4
**Confidence:** 3

**Summary:**

The authors adapt a visual transformer to be less sensitive to translations. Specifically, they adapt downsampling components, non-linearities and the attention mechanism. The authors build on existing work in this field, but combine various elements in a novel way. The resulting model is significantly more robust against certain standard and adversarial translations, while retaining a standard test accuracy that is mostly comparable with an adaptive polyphase sampling (APS) baseline. (See Table 1.) The most convincing results are in Fig 2, which compares the adversarial accuracy of different approaches on realistic shifts applied to Imagenet.

**Questions:**

1. Were you able to reproduce your findings on other datasets?
2. How exactly was HP tuning performed?

**Ethical Concerns:**

["NO or VERY MINOR ethics concerns only"]

**Final Justification:**

The paper proposes an adapted ViT that seems practical and useful, and addresses very small translations in images. Work is clearly presented and well motivated. Experimental results were extended during the review process and seem solid.

My only concern is the lack of hyperparameter (HP) tuning when comparing different approaches.  As the paper relies on its empirical results, I feel some additional analysis of the sensitivity of the approaches to HP tuning is required for the results to be fully trustworthy. Coupled with the error that slipped in (included in the main paper, found during the supplementary period), I am going to keep my score as a 4. I am still leaning towards accepting the paper, but with some reservations.

**Limitations:**

yes

**Quality:**

3

**Strengths And Weaknesses:**

Strengths:
- The paper proposes an adapted ViT that seems practical and useful.
- Work is clearly presented and well motivated.
- Experimental results seem solid (but also see weaknesses).

Weaknesses:
- For a practical paper that combines existing ideas carefully and methodically, I expected additional empirical results. All results are on Imagenet. Confirming results for additional datasets would have strengthened the paper.
- The sentence “We observe a slight improvement for the AF models 212 when training with a smaller batch-size; therefore, we reduce the batch-size from 1024 to 512 for the 213 AF-versions” seems to indicate that some form of hyperparameter tuning was performed. It is not clear what was done and which measures were taken to ensure a fair comparison across the different approaches.

---

> ### Author Rebuttal · Authors · 2025-07-30
>
> We thank the reviewer for their endorsement and constructive feedback. We are happy they found our work novel, practical and useful, well-motivated, clearly presented, and supported by solid experimental results.
> In the following, we address the questions that have been raised. We will incorporate these clarifications and additional results into the paper (or its appendix).
>
> > Were you able to reproduce your findings on other datasets?
>
> We evaluated our model solely on ImageNet-1K, as this is the only classification benchmark reported in our baseline model paper (XCiT [A]). This allowed us to avoid expensive hyperparameter tuning. In [A], the authors report results on additional detection and segmentation datasets, but these require training Mask R-CNN, which includes modules that are not trivial to convert to be fractional-shift equivariant under our framework (e.g., RoI Align, which uses bilinear interpolation). Therefore, we leave this as an interesting direction for future work and focus this paper on classification tasks.
>
> In response to the reviewer’s request, we evaluate our model on additional classification datasets. We fine-tune the models reported in the paper (pre-trained on ImageNet-1K) following the regime of [B]. Note that the ImageNet pre-training regimes of [A] and [B] are identical. For each dataset, we report the accuracy, integer shift consistency, and half-pixel shift consistency, as in the left part of Table 1 in the paper. Although the hyperparameters may not be optimal, the AF models maintain competitive accuracy while exhibiting near-perfect shift consistency. Note that we do not report results for the XCiT APS models, as they diverged with the current hyperparameters.
>
> | Model \ Dataset           | | CIFAR-10   |             |  | CIFAR-100  |         |  | Stanford Cars  |  |
> |---------------------------| :-:|:-------------:|:---------:|:--:|:-------------:|:-------:|:-:|:------------------:|:--:|
> | | Accuracy | Int. shift-consistency  | Half-pixel shift consistency | Accuracy | Int. shift-consistency  | Half-pixel shift consistency | Accuracy | Int. shift-consistency  | Half-pixel shift consistency |
> |XCiT-Nano (Baseline)  | 98.2 | 98.0 | 98.0     | 85.5 | 88.3 | 88.1  | 92.4 | 94.9 | 95.2|
> |XCiT-Nano-AF (Ours)  | 97.5 | 99.9 | 99.9     | 85.1 | 99.3 | 99.3 | 92.1 | 99.8 | 99.7|
> ||||||
> |XCiT-Small (Baseline)| 98.8 | 98.7 | 98.8      | 88.9 | 88.0 | 90.0  | 92.7 | 94.9 | 96.2 |
> |XCiT-Small-AF (Ours) | 98.6 | 100.0 | 100.0 | 87.1 | 99.7 | 99.6  | 92.4 | 99.8 | 99.7 |
>
>
> > How exactly was HP tuning performed?
>
> We refrained from performing full hyperparameter tuning due to limited resources. Thus, we retained the same hyperparameters as the XCiT paper (reported in Appendix C), with the following exception:
> We observed instability in training the AF models, which was resolved by reducing the batch size (and consequently the learning rate, scaled with the same ratio, as in [A] and [B]). We trained all Nano models (Baseline, AF, and APS) using batch sizes of 1024 and 512, and chose the best-performing configuration for each. We then used the same configuration for the Small versions of each model (i.e., 1024 for the baseline and 512 for APS and AF). Below, we report the accuracies for each of the Nano models under these batch sizes.
>
> | Model \ Batch Size  | 512    | 1024 |
> |---------------------|--------|------|
> |XCiT-Nano (Baseline) | 70.1   | 70.6 |
> |XCiT-Nano APS        | 68.7   | 67.2 |
> |XCiT-Nano-AF (Ours)  | 70.4   | 70.1 |
>
> ---
>
> [A] XCiT: Cross-Covariance Image Transformers, NeurIPS 2021.
>
> [B] Training data-efficient image transformers & distillation through attention, ICML 2021.

---

> > ### Comment · Reviewer_he7X · 2025-08-07
> >
> > Thank you for the update and additional results.
> >
> > Two follow-up questions:
> > 1. In your paper, you are comparing your results to APS (as an alternative technique to ensure shift invariance) rather than to the baseline only. Do you have comparable APS results for the additional datasets (CIFAR, Stanford Cars)?
> > 2. Some of your results when comparing the APS-enhanced baseline and the new architecture are quite close. Even if not done for all models, can you quantify the implications of tuning the key HPs for one or two of these models?
> >
> > (I am considering increasing my score.)

---

> > > ### Author Response · Authors · 2025-08-07
> > >
> > > We thank the reviewer for the additional questions and for considering increasing their rating.
> > >
> > > > In your paper, you are comparing your results to APS (as an alternative technique to ensure shift invariance) rather than to the baseline only. Do you have comparable APS results for the additional datasets (CIFAR, Stanford Cars)?
> > >
> > > In our initial rebuttal, we used for fine-tuning hyperparameters from [B], as reported in a discussion in their official implementation repository:
> > > * Image size: 224
> > > * Batch size: 512
> > > * Learning rate: 0.01
> > > * Optimizer: SGD
> > > * Weight-decay: 1e-4
> > > * Epochs: 1000.
> > >
> > > As mentioned in the rebuttal, **the APS models diverged under this setting**, and therefore, we have not reported their results for the additional datasets.
> > >
> > > To mitigate this, we re-ran fine-tuning of all the XCiT-Nano models with the ImageNet hyperparameters we originally used in the paper (changing only the number of epochs):
> > > * Image size: 224
> > > * Batch size: 512
> > > * Learning rate: 5e-4
> > > * Optimizer: AdamW
> > > * Weight-decay: 0.05
> > > * Epochs: 1000.
> > >
> > > With these hyperparameters, APS converged, but still performed worse than the other models. Note that here AF matches or slightly outperforms the baseline's accuracy across all datasets:
> > >
> > > | Model \ Dataset           | | CIFAR-10   |             |  | CIFAR-100  |         |  | Stanford Cars  |  |
> > > |---------------------------| :-:|:-------------:|:---------:|:--:|:-------------:|:-------:|:-:|:------------------:|:--:|
> > > | | Accuracy | Int. shift-consistency  | Half-pixel shift consistency | Accuracy | Int. shift-consistency  | Half-pixel shift consistency | Accuracy | Int. shift-consistency  | Half-pixel shift consistency |
> > > |XCiT-Nano (Baseline)  | 97.9 | 97.9  | 97.8 |  84.2 | 88.2   | 88.1 | 92.5 | 95.4  | 95.2
> > > |XCiT-Nano APS           | 97.6 | 100.0| 99.2 |  83.3 | 100.0 | 96.8 | 90.2 | 100.0| 95.1
> > > |XCiT-Nano-AF (Ours)  | 97.9 | 99.9  | 99.9 |  84.5 | 99.2   | 99.1 | 92.8 | 99.8  | 99.6
> > >
> > > > Some of your results when comparing the APS-enhanced baseline and the new architecture are quite close. Even if not done for all models, can you quantify the implications of tuning the key HPs for one or two of these models?
> > >
> > > Unfortunately, we will not be able to conduct full training sweeps by the end of the discussion period. However, since the reviewer has mentioned the similarity of the results of the APS and AF models, we provide a clarification about the results of APS (which has been given in the rebuttal to other reviewers who have raised concerns about this).
> > >
> > > *Table 1 and Figure 2 had an issue which gave APS an unfair advantage, but we already corrected this issue in Section B of the SM (Table 4 and Figure 3)*.
> > > Specifically, upon the main paper submission, we detected an error in our implementation of the APS model (in the results presented in Table 1 and Figure 2), leading to training with an effective patch size of 8 instead of 16, as was used in the Baseline and AF models. This led to better accuracy for the APS model and an unfair comparison vs. the other methods.
> > >
> > > The corrected results in Table 4 (in the SM) show that the APS model has lower accuracy than the AF model (APS 68.7 vs. AF 70.6 in the Nano versions and APS 80.1 vs. AF 80.7 in the Small versions). Consequently, the APS and AF models have similar adversarial accuracy under the integer grid (APS 68.7 vs. AF 68.9 in the Nano versions and APS 80.1 vs. AF 80.0 in the Small versions). Notably, the AF model has better adversarial accuracy under the half-pixel grid, with a larger margin than reported in Table 1 (APS 62.9 vs. AF 69.2 in the Nano versions and APS 76.0 vs. AF 79.9 in the Small versions). Additionally, the corrected results in Figure 3 demonstrate that the AF model outperforms the APS model on practical translations by a larger margin.
> > >
> > > Worth stressing again that we used the training regime reported in the Baseline [A] and APS [C] papers (where in the latter they used the same hyperparameters as the baseline for each architecture). We believe that, given that we have such a gap between AF and APS despite not doing significant hyperparameter tuning, can also be seen as a strength of our method (since further HP tuning may have improved the AFs results, but we did not need that).
> > >
> > > ---
> > >
> > > [A] XCiT: Cross-Covariance Image Transformers, NeurIPS 2021.
> > >
> > > [B] Training data-efficient image transformers & distillation through attention, ICML 2021.
> > >
> > > [C] Making Vision Transformers Truly Shift-Equivariant. CVPR 2024

---

> > > > ### Comment · Reviewer_he7X · 2025-08-09
> > > >
> > > > The updated results are useful! If the paper is accepted, I encourage the authors to include these in the main paper. I appreciate the difficulty of running additional HP sweeps, but feel some additional analysis is required for the results to be fully trusted. Coupled with the error that slipped into the main paper (found during the supplementary period) am going to keep my score as a 4. I am still leaning towards accepting the paper, but with some reservations.

---

### Decision · Program_Chairs · 2025-09-17

**Decision:**

Accept (poster)

**Comment:**

This paper presents an alias-free Vision Transformer (AF-ViT) designed for invariance to both integer and fractional image translations. Through targeted architectural modifications, the authors empirically demonstrate that their model achieves superior robustness to fractional shifts while maintaining competitive accuracy.

The paper's primary strength is its novel and well-motivated focus on achieving fractional shift invariance in ViTs, a significant and previously underexplored problem. The proposed method is technically sound and systematically constructed.

Initial weaknesses included limited empirical evaluation and concerns regarding fair hyperparameter tuning, which cast some doubt on the robustness of the results compared to baselines.

The paper is accepted because it makes a solid contribution to a novel problem. The authors comprehensively addressed all major concerns during the rebuttal by providing additional experiments and clarifications, ultimately presenting a convincing and well-validated method.